# OpenReview forum: "Training with Honeypots: Reshaping How LLMs Fail Under Adversarial Attacks"
_ICML.cc/2026/Conference — ICML 2026 regular_

### Official Review · Reviewer_FZRi · 2026-03-12

**Soundness:** 3
**Presentation:** 3
**Significance:** 3
**Originality:** 3
**Overall Recommendation:** 3
**Confidence:** 3

**Summary:**

This paper defined a training-time defense that reshapes the failure distribution of large language models under adversarial attack. Rather than treating all jailbreak outcomes as equally harmful, it constructs honeypot responses, which outputs that appear harmful to automated safety judges but lack real-world operational value, and use them as hard negatives in a DPO-style preference objective. The method is instantiated on top of both a simple threshold-based defense and Circuit Breakers, and evaluated across five automated judges, two model families, and multiple attack classes including an adaptive GRPO-based attacker. The results are generally positive: ASR drops by 10–21 points on average, the outputs that do get through are less actionable, and benign performance stays intact.

**Compliance With Llm Reviewing Policy:**

Affirmed.

**Key Questions For Authors:**

●	Why wasn't human evaluation conducted on the honeypot responses and actionability judgments?
●	How can practitioners deploy this method when it fails under hyperparameter transfer?
●	Would the method's effectiveness hold for larger models?

**Strengths And Weaknesses:**

Strength:

●	The proposed question is interesting and important. The observation that not all jailbreak successes are equally dangerous and that defenses should shape failure quality rather than only failure frequency is conceptually compelling and underexplored
●	The experiment applied to 5 different judges is more comprehensive than relying on a single judge.
●	Demonstrating that the approach works on top of both a simple threshold defense and a more complex representation-engineering defense (Circuit Breakers) is valuable for practitioners.

Weaknesses:
●	My major concern is while the honeypot responses are generated by prompting an uncensored Qwen-3 8B model with detailed instructions, it is better to contain human evaluation of the generated honeypots. The concern for the actionability judge is itself an LLM, and no ground-truth human assessment of actionability is provided for the training set.
●	Table 2 shows Circuit Breaking + Honeypot on Llama-3 (GRPO) actually increases ensemble ASR by 7.2 points. While the authors attribute this to a hyperparameter transfer issue, this is a major failure for robustness of the method. If the method requires attack-specific tuning to work, it's impractical for real deployment where future attacks are unknown.
●	Majority of the experiments are done on 8B models, the paper could conduct the experiment with a larger model size will be preferable.
●	Minor: "judge-detected violations" and "ASR" are used interchangeably sometimes but they're subtly different depending on whether the judge outputs binary labels or continuous scores.

---

> ### Author Rebuttal · Authors · 2026-03-31
>
> We thank the reviewer for their thoughtful feedback. Below, we address the questions they raised.
>
> > W1: No ground-truth human assessment of actionability is provided for the training set. Q1: Why wasn't human evaluation conducted on the honeypot responses and actionability judgments?
>
> For the rebuttal, we conducted a small-scale human expert judgment evaluation with 6 AI safety researchers from five different countries, all external to the author list. The annotators rated samples from the harmful dataset, the honeypot dataset, and jailbreak outputs from Qwen3-8B on 0–4 Likert scales for actionability (operational utility), on-topicness, and nonsensicalness. We plan to extend this into a large and systematic expert annotation study after the rebuttal process. The results are shown [here](https://anonymous.4open.science/r/icml_rebuttal_figures-AE75/README.md), along with a screenshot of the GUI tool developed for the annotation process.
>
> The results on the anonymized link demonstrate the utility of our method: honeypot responses are much more nonsensical (1.47), and much less actionable (1.50) than ground truth harmful responses (3.64, 3.38, respectively). In addition, models defended using honeypots have much lower actionability, going as low as 0.836 for the circuit-breaking honeypotted variant of Qwen.
>
> > W2, Q2: Table 2 shows Circuit Breaking + Honeypot on Llama-3 (GRPO) actually increases ensemble ASR by 7.2 points. While the authors attribute this to a hyperparameter transfer issue, this is a major failure for robustness of the method. If the method requires attack-specific tuning to work, it's impractical for real deployment where future attacks are unknown.
>
> We note that the GRPO results on Llama-3 are based on a single configuration. We will train and evaluate additional defensive hyperparameters for the Llama-3 model after the rebuttal process. While we observe an increase in one specific setting, for embedding-based attacks we test multiple settings (6 per model), where honeypot regularization consistently yields improvements across the tested hyperparameters. This suggests the observed degradation is not representative of the overall method. Additional results with Qwen3-14B further support this: we observe near-zero (0%) ASR under GPT-OSS.
>
> More broadly, sufficiently strong adaptive attacks such as GRPO have been shown to break most defenses given enough optimization [1]. In this setting, the key question is how the model behaves under attack. Our human study shows that models trained with honeypots produce significantly less actionable jailbreak outputs under GRPO, with actionability dropping as low as 0.836 for the circuit-breaking + honeypot variant of Qwen3-8B.
>
> > W3, Q3: Majority of the experiments are done on 8B models, the paper could conduct the experiment with a larger model size will be preferable.
>
> In the rebuttal, we include additional results on Qwen3-14B, where our method achieves 0% measured ASR under GPT-OSS, with no loss of benign capability. These results provide evidence that honeypot regularization can be effective with larger models.
>
> > W4: Minor: "judge-detected violations" and "ASR" are used interchangeably sometimes but they're subtly different depending on whether the judge outputs binary labels or continuous scores.
>
> We thank the reviewer for the comment and will clarify the terms in our latest manuscript.
>
> In addition, we invite the reviewer to consult the rest of our [supplementary rebuttal experiments](https://anonymous.4open.science/r/icml_rebuttal_figures-AE75), which include an ablation study of the loss terms that clarifies the role of each optimization component. We believe these results greatly strengthen the significance of our results, and improve the overall soundness of the paper.
>
> [1]: Nasr et al. (2025). The attacker moves second: Stronger adaptive attacks bypass defenses against LLM jailbreaks and prompt injections.

---

> > ### Author Rebuttal · Reviewer_FZRi · 2026-04-02
> >
> > Most of my concerns have been adequately addressed.

---

> > > ### Author Response · Authors · 2026-04-08
> > >
> > > We thank the reviewer for their effort in helping to improve the paper, and are glad to hear most concerns have been addressed.
> > >
> > > We would like to briefly clarify the remaining point regarding the degradation observed for Circuit Breakers combined with DPO on Llama-3 under GRPO. The Circuit Breaker objective encourages disruption of internal representations, leading to nonsensical logits on harmful inputs, but the DPO objective operates on logits and enforces a preference for honeypots, which implicitly encourages coherence. When the DPO signal is too strong, these objectives can partially conflict. In particular, the hyperparameter selection on embedding attacks found that a strong DPO setting is preferable for Llama for that setting (beta=5, margin=1), and a moderate one is preferable for Qwen (beta=2, margin=0.5). We believe this may have contributed to the slightly worse performance of the Llama model under GRPO attacks. Prior work has shown that circuit breaking’s effectiveness depends on both hyperparameter choices and the attack setting [1].
> > >
> > > For the final version, we will include additional post-rebuttal experiments with a larger hyperparameter search DPO settings. Existing circuit breaking inspired papers have seen success with disrupting representation of late layers [2] and we plan to imitate their approach.
> > > We also note that, beyond this specific setting, our results remain consistent across multiple configurations. Additionally, on Qwen3-14B we observe near-zero measured ASR under GPT-OSS with no degradation in benign capability. Our human evaluation further confirms that, even under strong adaptive attacks, the outputs produced by honeypot-trained models are less actionable.
> > >
> > > We hope we were able to answer all of the reviewer’s concerns. If there are remaining questions or points that we did not fully clarify, we would greatly appreciate further clarification so we can better address them in the final version. We kindly invite the reviewer to consider our additional comments and, if our rebuttal sufficiently addresses the reviewer's concerns, we would be grateful for a reconsideration of the score.
> > >
> > > [1] Schwinn et al. (2024) Revisiting the robust alignment of circuit breakers.
> > >
> > > [2]: Yousefpour et al. (2025). Representation bending for large language model safety.

---

### Official Review · Reviewer_yReW · 2026-03-12

**Soundness:** 3
**Presentation:** 3
**Significance:** 3
**Originality:** 2
**Overall Recommendation:** 5
**Confidence:** 3

**Summary:**

The authors propose using honeypot-defenses to reshape LLM responses on harmful requests to be inactionable, and find that honeypot-defenses also improve robustness in several attack settings. The authors evaluated the method and baselines with both target embedding attacks and GRPO adaptive attacker, with a comprehensive suite of automatic judges. The author found that honey regularization substantially improves robustness to automatic jailbreaks, and that honeypot regularization reduces actionable harm. The utility of the method is also demonstrated through the preservation of scores on benign tasks.

**Compliance With Llm Reviewing Policy:**

Affirmed.

**Final Justification:**

The authors adequately addressed my questions, and the rebuttal experiments with the human annotation, larger model, and ablation further supported that the method provides meaningfully less actionable responses.

**Key Questions For Authors:**

- Why do you think honeypot regularization improves robustness to automatic jailbreaks?
- Why guide honeypot response construction to not admit its lack of actionability "Don't mention anywhere that you are not actually helpful."? This is not related to the main goal of the paper, but I'm curious why this was explicitly in your prompt, as it may be good to clearly distinguish honeypot responses versus non-honey responses.
- Were any experiments of few-shot fine-tune attack done as another comparison between honeypot method and baseline methods?

**Limitations:**

yes

**Strengths And Weaknesses:**

Strengths
- The paper is clearly presented, with good motivation and benefits of the method proposed.
- The paper presents an interesting idea of including a weaker failure case of the honeypot, and provides evidence experimentally that it is able to improve robustness and decrease actionability.
- The paper include multiple judges and adaptive attacks demonstrating that the usefulness of the method is not judge or attack specific.
- It is interesting that honeypot regularization improves robustness to automatic jailbreaks

Weakness
- Related work around LLM failing less severely: https://openreview.net/forum?id=XdRv6I80L1
- In 6.1, it is suggested that honey point regularization targets deeper failure modes than simple refusal triggering. Is there more evidence on this suggestion?

---

> ### Author Rebuttal · Authors · 2026-03-31
>
> We thank the reviewer for his thoughtful and positive feedback. Below, we address the questions they raised.
>
> We invite the reviewer to consult our [supplementary rebuttal experiments](https://anonymous.4open.science/r/icml_rebuttal_figures-AE75/README.md), which include:
> - R1: a manual expert evaluation of the actionability of the training datasets and resulting jailbreaks;
> - R2: an ablation study of the loss terms, which clarifies the role of each optimization component;
> - R3: additional results on larger models, which provide evidence on scalability.
>
> We believe these results strengthen the significance of our results, and improve the overall soundness of the paper.
>
> > W1: Related work around LLM failing less severely
>
> We thank the reviewer for the relevant reference, and we will add it in our related work section.
> While both approaches aim to reduce the utility of harmful outputs, our method differs in that we explicitly train on honeypot responses, which all share similarly low actionability, rather than training along a spectrum of actionable answers. In addition, we show our honeypot dataset is perceived as highly harmful by most existing judge classifiers used in academia. We apply our approach to strong, widely used defenses such as circuit breaking and evaluate our defenses against more challenging adaptive attacks (GRPO).
>
> > W2: In 6.1, it is suggested that honey point regularization targets deeper failure modes than simple refusal triggering. Is there more evidence on this suggestion?
>
> In our new rebuttal human evaluation, honeypot-trained models exhibit substantially lower actionability while maintaining comparable or only moderately increased refusal rates. This indicates that the effect is not only driven by refusal triggering. Across our evaluations, we also observe that honeypot regularization has a stronger impact on actionability-based metrics (e.g., GPT-OSS), even when refusal-based judges (e.g., WildGuard) remain relatively high.
>
> > Why do you think honeypot regularization improves robustness to automatic jailbreaks?
>
> Honeypot training shifts the model toward the more judge-inconsistent failure modes and away from a single clear harmful mode (honeypot responses exhibit markedly lower cross-judge agreement than genuinely harmful responses) Therefore, even when an attacker optimizes a single judge such as StrongREJECT, the best reachable responses under the defended model can still receive lower StrongREJECT scores, since they are less complete and less consistent with the harmful target pattern that the judge rewards.
>
> > Q1: Why guide honeypot response construction to not admit its lack of actionability "Don't mention anywhere that you are not actually helpful."? This is not related to the main goal of the paper, but I'm curious why this was explicitly in your prompt, as it may be good to clearly distinguish honeypot responses versus non-honey responses.
>
> We guide the honeypot responses to not admit actionability, as we observed that without this, the model often eluded to it (e.g, it started with "Sure, here is a non-operational answer.."). Such signals make the responses easy to distinguish, even for lightweight judges such as the StrongREJECT judge, and could potentially be exploited by adaptive attackers.
>
> > Were any experiments of few-shot fine-tune attack done as another comparison between honeypot method and baseline methods?
>
> In the appendix, Table 6 reports results under a harmful prefix-tuning attack after fine-tuning on 5000 harmful prompt-response pairs. While this is not exactly a few-shot fine-tuning attack, it is a closely related tampering setting. In that experiment, averaged across three defense hyperparameter settings, honeypot regularization does not improve the tamper-resistance of the underlying defenses (with a ensemble ASR reduction of 0.75% compared to the non-honeypot baselines, averaged over all configurations and models tested). The circuit-breaking defense has been shown to be quite brittle under finetuning attacks [1]: we leave the extension of the honeypot regularization approach to tamper-resistant defenses such as TAR or Vaccine to future work.
>
> [1]: Hossain et al (2026). TamperBench: Systematically Stress-Testing LLM Safety Under Fine-Tuning and Tampering.

---

> > ### Author Rebuttal · Reviewer_yReW · 2026-04-03
> >
> > The rebuttals have addressed my questions and I have raised my score.

---

> > > ### Author Response · Authors · 2026-04-08
> > >
> > > We sincerely thank the reviewer for their score adjustment and constructive feedback, which helped strengthen our paper.

---

### Official Review · Reviewer_zRQw · 2026-03-12

**Soundness:** 2
**Presentation:** 2
**Significance:** 3
**Originality:** 3
**Overall Recommendation:** 3
**Confidence:** 3

**Summary:**

This paper proposes a dataset construction and training procedure to improve LLM safety. Specifically, it first constructs a dataset of “honeypot” responses — compliant responses to harmful queries that lack actionable or truly harmful information — and then trains an LLM to prefer these responses over their actionable counterparts. Through DPO-style fine-tuning and integration with an existing representation-engineering defense, the paper demonstrates that honeypot training can cause models to defer to non-actionable responses when initial alignment is compromised, thereby mitigating the downstream impact of several jailbreak attacks.

**Compliance With Llm Reviewing Policy:**

Affirmed.

**Final Justification:**

I remain overall concerned about the paper's discussion of actionability and its connection to robustness against prompt-based jailbreaks.

The paper's initial framing is to replace harmful responses with low-actionability "honeypots" that fool typical judges. However, the high actionability of the honeypot dataset (36.1%) is in tension with this framing. The authors then claim that a certain amount of actionability is needed to defend against attacks like GRPO. While the authors have provided additional experiments during the rebuttal period, I do not feel they offer sufficient evidence for this claim. In particular, the results report actionability conditioned on the HarmBench score, which does not measure the true underlying proportion of actionable responses. Additionally, the results are inconclusive: while increasing the actionability of the honeypot dataset decreases conditional actionability under GRPO for Llama, this does not hold true for Qwen, as Setting 1 actually achieves the lowest actionability.

More generally, this discussion appears to shift the paper's overall framing. Rather than simply replacing harmful responses with low-actionability outputs, a precise balance is needed to prevent exploitation via adversarial attacks. This is interesting, but the paper largely lacks any discussion of this topic, and the quantitative results provided in the rebuttal period are not convincing.

In conjunction with my original concerns (that honeypot training does not consistently improve robustness and that actionability is not well controlled during dataset generation), I feel this paper needs a more careful methodology and framing that is non-trivial to address in a revision period. I therefore retain my score.

**Key Questions For Authors:**

1. Can the authors further justify or analyze the construction of the honeypot dataset? In particular, I wonder if an additional filtering step would help reduce its actionability and thus effectiveness in downstream training.

2. Can the authors provide some analysis on why honeypot training can increase ASR under certain conditions? Diagnosing the root cause would provide valuable guidance for future work or for refining the proposed method.

3. Can the authors ablate the impact of each optimization objective introduced in Section 4 and Appendix A.1? This would help justify each component's inclusion, making the method design more rigorous.

**Limitations:**

Yes

**Strengths And Weaknesses:**

### **Strengths**

1. The paper is well-motivated, as existing defenses typically treat jailbreaks as a binary outcome (a refusal or a compliant response), without considering the real-world harms or the actionability of the generated outputs. Assessing whether models can be trained to prefer non-actionable, honeypot-style responses therefore offers a practical way to mitigate harmful generations without strictly enforcing refusal behavior.

2. The definition and evaluation of “actionability” provide a new perspective on quantifying harmful LLM generations, which complements existing ASR metrics.

3. The proposed training methods generally reduce actionability, demonstrating the feasibility of reshaping safety alignment through honeypot responses.

### **Weaknesses**

1. The honeypot dataset is a critical component of the proposed defense (it is listed as the first contribution), but it appears trivially constructed without much justification or analysis. Furthermore, the honeypot dataset responses still have high actionability (36.1%), suggesting the proposed dataset generation strategy is insufficient for producing high-quality outputs.

2. There are a significant number of instances in Table 1 and Table 6 where training with honeypot responses increases ASR, even under the actionability judge GPT-OSS, limiting the method’s purported effectiveness.

3. The method contains many different optimization objectives (the final equation in Appendix A.1 has 5 terms) without a clear justification of why they are needed. This complexity almost certainly adds runtime and memory overhead to training, so I would expect a more rigorous evaluation of each component's impact.

4. The method and experimental setup omit several details that are important for interpretability and reproducibility. For example:

- The input prompt symbol $x$ is used across multiple training objectives without clarifying which distribution it is sampled from.
- The cross-entropy loss $CE_\theta(x)$ is defined without specifying the target labels or target responses.
- To my understanding, neither the benign data distribution nor the dataset of harmful responses used for evaluation is clearly described.

More generally, the notation is difficult to follow, and some symbols appear inconsistent across sections (e.g., $\lambda$ is used for weighting terms in the main text, whereas Appendix A.1 switches to $w$). These issues may be addressable during rebuttal, but in their current form, they raise concerns about the paper's presentation and methodological clarity.

---

> ### Author Rebuttal · Authors · 2026-03-31
>
> We thank the reviewer for their thoughtful feedback. Below, we address the questions they raised.
>
> We invite the reviewer to consult the figures of our [rebuttal experiments](https://anonymous.4open.science/r/icml_rebuttal_figures-AE75/README.md), which include:
> - R1: a manual expert evaluation of the actionability of the training datasets and resulting jailbreaks;
> - R2: an ablation study of the loss terms, which clarifies the role of each optimization component;
> - R3: additional results on larger models (Qwen3-14B), which provide evidence on scalability.
>
> We believe these results greatly strengthen the significance of our results, and improve the overall soundness of the paper.
>
> > W1, Q1: The honeypot dataset [...] appears trivially constructed [...]. Can the authors further justify or analyze the construction of the honeypot dataset?
>
> The honeypot generation process is intentionally simple: our goal was to create a scalable way to generate responses that are hard for judges to distinguish from true harmful outputs, but that are also natural for the model to respond. The honeypot replies produced by our data generation pipeline reflect what the model itself would plausibly say if it were jailbroken (i.e behaving like an uncensored model), but with an additional incentive introduced via a system prompt that discourages operational and actionable content. This results in honeypot responses that closely match the model’s generative tendencies.
>
>  > Furthermore, the honeypot dataset responses still have high actionability (36.1%), suggesting the proposed dataset generation strategy is insufficient for producing high-quality outputs.
>
> Our objective is to make them less operationally useful than true harmful responses while still keeping them close enough that existing judges often treat them similarly. If the honeypots were pushed to near-zero actionability under GPT-OSS, this could indicate that they had become easy decoys and could be easy for an adaptive attack like GRPO to distinguish.
>
> For the rebuttal, we conducted a small-scale human expert judgment evaluation with 6 AI safety researchers from five different countries, all external to the author list. The annotators rated samples from the harmful dataset, the honeypot dataset, and jailbreak outputs from Qwen3-8B on 0–4 Likert scales for actionability (operational utility), on-topicness, and nonsensicalness. We plan to extend this into a large and systematic expert annotation study after the rebuttal process.
>
> Our findings indicate that honeypot responses are much more nonsensical (1.47), and much less actionable (1.50) than ground truth harmful responses (3.64, 3.38, respectively). In addition, models defended using honeypots have much lower actionability, as low as 0.836 for the circuit-breaking honeypotted variant of Qwen.
>
> > W2: There are a significant number of instances in Table 1 and Table 6 where training with honeypot responses increases ASR, even under the actionability judge GPT-OSS, limiting the method’s purported effectiveness.
>
> We note that the GRPO results on Llama-3 are based on a single configuration. We will train and evaluate additional defensive hyperparameters for the Llama-3 model after the rebuttal process. While we observe an increase in one specific setting, for embedding-based attacks we test multiple settings (6 per model), where honeypot regularization consistently yields improvements across the tested hyperparameters. This suggests the observed degradation is not representative of the overall method. Additional rebuttal results with Qwen3-14B further support this: we observe near-zero (0%) ASR under GPT-OSS.
>
> > W3, Q3: Can the authors ablate the impact of each optimization objective introduced in Section 4 and Appendix A.1?
>
> We performed an ablation study of the loss terms on Qwen 3 8B and Llama 3 8B for the threshold defense. We ablate each term as described by the reviewer. Our [results](https://anonymous.4open.science/r/icml_rebuttal_figures-AE75/README.md) show that overall, our loss terms are needed to keep a low ASR across both evaluated models. In particular, removing the honeypot CE term pushes the preference learning signal to massively prefer honeypots, leading to a more harmful mode. Removing the KL divergence term yields lower scores (GSM8K: 88% compared to 92%).
>
> > W4: The method and experimental setup omit several details
>
> We thank the reviewer for their valuable comments. We used the same experimental setup as the circuit-breaking paper [1]: Data used for the benign and cross-entropy terms is drawn from UltraChat [2], harmful data is drawn from the circuit-breaking dataset [1], and harmful behaviors for evaluation are taken from the HarmBench validation set. We will clarify all details in our latest manuscript.
>
> [1]: Zou, A. (2024). Improving alignment and robustness with circuit breakers
>
> [2]: Ding, N. (2023). Enhancing chat language models by scaling high-quality instructional conversations

---

> > ### Author Rebuttal · Reviewer_zRQw · 2026-04-03
> >
> > I thank the authors for their responses and find that most of my concerns have been addressed.
> >
> > However, I am not fully convinced by the justification of the honeypot dataset's high actionability. It seems the authors argue that constructing a lower-actionability dataset would be problematic because an adaptive GRPO-based attacker could recognize such outputs and bypass them, but they offer no empirical evidence. Have the authors actually tested this?
> >
> > More generally, if greater efforts are taken to reduce the actionability of the honeypot dataset, does it improve or decrease the resulting robustness of trained models?

---

> > > ### Author Response · Authors · 2026-04-08
> > >
> > > We thank the reviewer for their effort in helping to improve the paper, and are glad to hear most concerns have been addressed.
> > >
> > > > It seems the authors argue that constructing a lower-actionability dataset would be problematic because an adaptive GRPO-based attacker could recognize such outputs and bypass them, but they offer no empirical evidence. Have the authors actually tested this? More generally, if greater efforts are taken to reduce the actionability of the honeypot dataset, does it improve or decrease the resulting robustness of trained models?
> > >
> > > Our original honeypot dataset was less actionable than the current one, but was also considered less harmful by other judges. All judges partially account for actionability: for instance, a harmful but extremely nonsensical response would tend to receive a lower StrongREJECT score as a more actionable one. Since the GRPO attack relies on an LLM-as-a-judge reward signal (correlated with actionability) the attack is implicitly guided toward producing more actionable responses, more typically aligned with what the LLM-as-a-judge considers to be harmful. This behavior was already observed during our initial development of the honeypot dataset, where very low actionability variants that sacrifice standard judge harmfulness were less effective.
> > >
> > > To further investigate this effect and address the reviewer’s question, we include an [additional rebuttal experiment (R4)](https://anonymous.4open.science/r/icml_rebuttal_figures_2-AFD3/README.md), which we invite the reviewer to consult.
> > >
> > > To quantify the effect of the actionability of the honeypot training dataset, we construct four datasets (Settings 0, 1, 2, and 3) with progressively increasing actionability by [adding instructions in our system prompt](https://anonymous.4open.science/r/icml_rebuttal_figures_2-AFD3/r4_actionability_study/system_prompt.md). Each dataset consists of 3000 examples.  We train defenses on 8B versions of Qwen3 and Llama3 on each dataset variant with the Threshold defense, and evaluate robustness under adaptive attacks and embedding-space attacks. The measured actionability ranges from *0.18* (Setting 0) to *0.7* (Setting 3).
> > >
> > >
> > > Our main findings can be summarized as follows:
> > >
> > > - Generating less actionable honeypots tends to result in lower honeypot dataset harmfulness scores for other judges (setting 0: 0.60 StrongREJECT score compared to 0.85 for setting 3)
> > >
> > > - GRPO attack successes are much more likely to be actionable than embedding attack successes: for soft prompts, among HarmBench-positive outputs, only *41.3%* are GPT-OSS actionable, compared to *71.4%* for GRPO.
> > >
> > > - The conditional probability that HarmBench-positive outputs are actionable (P(GPT-OSS=1 | HarmBench=1)) decreases as the honeypot dataset gets more actionable: for example, *100%* of Llama Setting 0 GRPO jailbreaks are actionable, compared to under *50%* for Setting 3 ([conditional_asr.png](https://anonymous.4open.science/r/icml_rebuttal_figures_2-AFD3/r4_actionability_study/conditional_asr.png))
> > >
> > > - An additional sentence embedding analysis showed that as actionability increases, honeypot responses become progressively closer to the original harmful data ([sentence_embedding.png](https://anonymous.4open.science/r/icml_rebuttal_figures_2-AFD3/r4_actionability_study/sentence_embedding.png))
> > >
> > > Due to time constraints, this experiment used a reduced defense-training and GRPO budget, with fewer training steps (1000), a single hyperparameter choice, fewer evaluation behaviors (41), and lighter GRPO settings (we report the first successful break) than our full-scale runs. While these current results are already indicative, we take the reviewer’s concerns seriously and will include a more comprehensive version of this study in the final manuscript, where we will explicitly control for harmfulness across weaker judges (matching dataset ASR across settings).
> > >
> > > With the completed rebuttal experiments (R1 – R4), we hope we were able to answer all of the reviewer’s concerns. We kindly invite the reviewer to consider our additional comments and, if our rebuttal sufficiently addresses the reviewer's concerns, we would be grateful for a reconsideration of the score.

---

### Official Review · Reviewer_dzFW · 2026-03-13

**Soundness:** 4
**Presentation:** 4
**Significance:** 3
**Originality:** 3
**Overall Recommendation:** 6
**Confidence:** 4

**Summary:**

This paper analyzes harmful response judges and demonstrates that LLM judges assign high harmfulness even to queries that have low operational utility. These queries are generated using an uncensored model and validated against the LLM judges to form a honeypot dataset. This honeypot dataset is used to finetune open source LLMs Llama-3 8B and Qwen-3 8B using a DPO finetuning process that reshapes preferences for harmful (according to the LLM judge) responses while also placing non-operational utility responses to be higher preference within this category of harmful responses. This honeypot response tuning when coupled with threshold filtering increases robustness in terms of Attack Success Rate (ASR) across both open source models. When coupled with Circuit Breakers though the combination decreases robustness in terms of ASR for Llama-3 8B when attacked by a GRPO adapative attacker. Although in all cases the addition of honeypot finetuning reduces the operational utitlity of harmful responses and preserves benign capabilities.

**Compliance With Llm Reviewing Policy:**

Affirmed.

**Final Justification:**

The paper initially showed strong performance but lacked human evaluations to corraborate automated judge decisions and didn't discuss effects on larger models. The author's rebuttal addressed my primary concerns about human judgement and scaling behavior making me update my score to a strong accept.

**Key Questions For Authors:**

Q.1 The operational utility of harmful responses is more likely to be a spectrum whereas the paper makes a clear binary distinction between operational and non-operational responses. Are there any cases where the GPT-OSS judge classifies a response as non-operational but the response is largely of operational utility?

**Limitations:**

Yes

**Strengths And Weaknesses:**

Strengths:
- Strong motivation and framing of harmful responses by differentiating between them based on utility of the responses
- Thorough evaluation setup using different types of LLM judges specializing judging harmful responses based on different criteria showing decreased ASR and operational utility of responses generated by successful attacks
- Honeypot finetuning largely preserves benign preferences (as measured by TruthfulQA) making the technique highly practically viable
- Well-written, easy to follow prose and solid presentation of figures and tables

Weaknesses:
- Honeypot queries are generated using a single model and doesn't take into account human judgement of these queries, validation of high harmfulness according to LLM judges is not sufficient to demonstrate the low utility of these queries
- Coverage of harmfulness categories is not discussed
- Models tested are both 8B parameters making the scaling behavior of the technique uncertain

---

> ### Author Rebuttal · Authors · 2026-03-31
>
> We thank the reviewer for his thoughtful and positive feedback. Below, we address the questions they raised.
>
> > W1: Honeypot queries [...] don’t take into account human judgment
>
> We conducted a small-scale human expert judgment evaluation with 6 AI safety researchers from five different countries, all external to the author list. The annotators rated samples from the harmful dataset, the honeypot dataset, and jailbreak outputs from Qwen3-8B on 0–4 Likert scales for actionability (operational utility), on-topicness, and nonsensicalness. We plan to extend this into a large and  systematic expert annotation study after the rebuttal process.
>
> The results, found on [this anonymized link](https://anonymous.4open.science/r/icml_rebuttal_figures-AE75), demonstrate the utility of our method: honeypot responses are much more nonsensical (1.47), and much less actionable (1.50) than ground truth harmful responses (3.64, 3.38, respectively). In addition, models defended using honeypots have much lower actionability, going as low as 0.836 for the circuit-breaking honeypotted variant of Qwen.
>
> > W2: Coverage of harmfulness categories is not discussed
>
> We use queries from the HarmBench evaluation suite, which contain 7 categories of behavior: Cybercrime & Unauthorized Intrusion, Chemical & Biological Weapons/Drugs, Copyright Violations, Misinformation & Disinformation, Harassment & Bullying, Illegal Activities, and General Harm. We will clarify this in our latest manuscript.
>
> > W3: Models tested are both 8B parameters making the scaling behavior of the technique uncertain
>
> To assess the scaling behavior of our method, we train both the threshold defense and the circuit breaking defense on Qwen3-14B. The results at [this anonymized link](https://anonymous.4open.science/r/icml_rebuttal_figures-AE75) show clear gains across tested judges using honeypots: in particular, the actionability judge measures **0%** ASR on both threshold and circuit breaking models, with no significant drop in benign capability.
>
> In addition, we invite the reviewer to consult the rest of our  [rebuttal experiments](https://anonymous.4open.science/r/icml_rebuttal_figures-AE75/README.md)
> , which also include an ablation study of the loss terms, clarifying the role of each component.

---

> > ### Author Rebuttal · Reviewer_dzFW · 2026-04-02
> >
> > I thank the authors for their extended experiments and given that my concerns are effectively resolved I will upgrade my score by 1 point.

---

> > > ### Author Response · Authors · 2026-04-08
> > >
> > > We sincerely thank the reviewer for their score adjustment and constructive feedback, which helped strengthen our paper.

---

### Decision · Program_Chairs · 2026-04-30

**Decision:**

Accept (regular)

**Comment:**

This paper proposes finetuning models to provide, in case of a compliant answer (i.e., a failure to provide a clear refusal), non-actionable answers (honeypots) to harmful queries. This improves the adversarial robustness of the safety model when measuring the actual actionability of the compliant answers.

Two reviewers were very positive about the paper, and two were more nuanced. The main concern that remained at the end of the discussion was that the honeypot dataset requires a certain level of actionability to be effective against GRPO attacks (which judges the answer's actionability when crafting the attack), which is an important nuance that goes against the intial idea conveyed in the abstract that the models should provide " low-utility, non-actionable outputs rather than highly actionable harm". Given the additional experiments provided by the author at the end of the discussion exploring honeypot datasets with different levels of actionability, and given the fact that all the other points have been addressed by the authors, I recommend accepting this paper.

I recommend that the authors adjust the framing of their story accordingly and thoroughly explore and discuss the balance between low and medium actionability required for the honeypot dataset.